# An RNA Metabolism and Surveillance Quartet in the Major Histocompatibility Complex

**DOI:** 10.3390/cells8091008

**Published:** 2019-08-30

**Authors:** Danlei Zhou, Michalea Lai, Aiqin Luo, Chack-Yung Yu

**Affiliations:** 1School of Life Science, Beijing Institute of Technology, Beijing 100081, China; 2The Abigail Wexner Research Institute at Nationwide Children’s Hospital, Columbus, OH 43205, USA; 3Department of Pediatrics, The Ohio State University, Columbus, OH 43205, USA

**Keywords:** DXO, DOM3Z, NELF-E, RD, SKIV2L, SKI2W, STK19, RP1, NSDK, RLR, miR1236, SVA, RNA quality control, 5′→3′ RNA decay, 3′→5′ mRNA turnover, antiviral immunity, interferon β, promoter-proximal transcriptional pause, exosomes, nuclear kinase, hepatocellular carcinoma, Ski complex, trichohepatoenteric syndrome, melanoma

## Abstract

At the central region of the mammalian major histocompatibility complex (MHC) is a complement gene cluster that codes for constituents of complement C3 convertases (C2, factor B and C4). Complement activation drives the humoral effector functions for immune response. Sandwiched between the genes for serine proteinase factor B and anchor protein C4 are four less known but critically important genes coding for essential functions related to metabolism and surveillance of RNA during the transcriptional and translational processes of gene expression. These four genes are NELF-E (RD), SKIV2L (SKI2W), DXO (DOM3Z) and STK19 (RP1 or G11) and dubbed as NSDK. NELF-E is the subunit E of negative elongation factor responsible for promoter proximal pause of transcription. SKIV2L is the RNA helicase for cytoplasmic exosomes responsible for degradation of de-polyadenylated mRNA and viral RNA. DXO is a powerful enzyme with pyro-phosphohydrolase activity towards 5′ triphosphorylated RNA, decapping and exoribonuclease activities of faulty nuclear RNA molecules. STK19 is a nuclear kinase that phosphorylates RNA-binding proteins during transcription. STK19 is also involved in DNA repair during active transcription and in nuclear signal transduction. The genetic, biochemical and functional properties for NSDK in the MHC largely stay as a secret for many immunologists. Here we briefly review the roles of (a) NELF-E on transcriptional pausing; (b) SKIV2L on turnover of deadenylated or expired RNA 3′→5′ through the Ski-exosome complex, and modulation of inflammatory response initiated by retinoic acid-inducible gene 1-like receptor (RLR) sensing of viral infections; (c) DXO on quality control of RNA integrity through recognition of 5′ caps and destruction of faulty adducts in 5′→3′ fashion; and (d) STK19 on nuclear protein phosphorylations. There is compelling evidence that a dysregulation or a deficiency of a NSDK gene would cause a malignant, immunologic or digestive disease.

## 1. Introduction

In 1984, Professor Rodney Porter’s team in Oxford reported the cloning and confirmed the physical linkage of the genes for human complement C4A and C4B, factor B and C2 through overlapping cosmids [1]. These genes are located at the central region of the human major histocompatibility complex (MHC), a region that is associated with numerous complex autoimmune diseases [2,3,4,5,6,7]. They code for the essential constituents of the classical, lectin and alternative pathways C3 convertases. The complement system is a major effector arm for both innate and adaptive immune systems [8]. While C2 and factor B are only separated by an intergenic distance of 421 bp [9], it was a bit puzzling that the genes for C4 and factor B are separated by a large gap of 30 kb. Thus, it was satisfying when a genomic DNA probe upstream of the C4 promoter region successfully fished out an 1.1 kb cDNA clone that was dubbed as RP (to memorialize the late Professor Rodney Porter) [10,11]. Working from the other end, Meo’s team reported the cloning of RD that is up to 149 bp tail-to-tail with factor B [12]. The protein sequence of RD is fascinating because of its 23 consecutive copies of positively charged arginine (R) and negatively charged aspartic acid (D), not to mention its leucine zipper and RNA binding motif. Between RD and RP, we found two more genes, SKI2W and DOM3Z [13,14,15].

Through biochemical and molecular biologic research to decipher their functions, these four “novel” genes turn out to be extremely old in evolution. Except for RP, structurally and functionally related genes and proteins can be traced all the way back to yeast or metazoans. The modern names for these four genes are NELF-E, SKIV2L, DXO and STK19 (NSDK), respectively. Unlike the Class I and Class II and complement that have distinct immunologic functions involving proteins, NSDK seems to play fundamental life functions in eukaryotes and actively engaged in the surveillance of RNA integrity including the 5′ cap and 3′ tail. In the process of cell metabolism, these four genes play critical roles on transcriptional control, RNA turnover and signaling. They may also fine-tune the innate immune response or type I interferon-mediated inflammation elicited by infections of RNA viruses. Here we give a brief account on the molecular genetics and biology of this RNA metabolism and surveillance “quartet” in the MHC (Figure 1, Table 1).

## 2. NELF-E or RD

### To Start a Long Journey, Get Prepared, Do a Test Run, Pause to Check Things Out, Modify and Let Go: The Amazing RD or Negative Elongation Factor Subunit E (NELF-E) in Gene Transcription

The human RD gene, also known as NELF-E and D6S45, is located between complement factor B (Bf) and the SKIV2L gene. This gene was discovered by Meo and colleagues in 1988 [12]. RD is organized in a tail-to-tail configuration with Bf (Figure 1). There are 205 nucleotides between the 3’ ends of RD and Bf. The 5′ regulatory region of the RD gene is rich in CpG sequences [17]. Transcripts for RD are ubiquitously expressed but the highest expression level is found in the testis.

The RD gene is 6.7 kb in size and consists of 11 exons. It encodes 380 amino acids. The calculated molecular weight of the RD protein is 46 kDa [17,18]. The RD protein contains a leucine-zipper motif at the N terminal region, which facilitates interactions with other proteins [12,17,19]. A striking feature of RD, however, is a tract of tandem repeats with positively and negatively charged amino acids, which are predominantly Arg-Asp (RD), from residues 184 to 243, which are encoded by exon 7. Close to the carboxyl terminal region is an RNA recognition motif (RRM), which is located between residues 264 and 327 and encoded by exons 8–10. Thus, RD belongs to ribonucleoprotein family. The high degree of sequence similarities between human and murine RD both at DNA and amino acid levels suggested functional importance [12,15,17,20].

One feature for the regulation of gene expression in metazoans is the promoter-proximal transcriptional pausing of RNA polymerase II. This phenomenon is carried out by two complexes known as the negative elongation factor (NELF) and DRB (5,6-dichloro-1-β-d-ribofuranosyl-benzimidazole) sensitivity inducing factor (DSIF). NELF acts cooperatively with DSIF to strongly represses transcriptional elongation by RNA polymerase II (RNAPII), about 25–50 nucleotides downstream of transcriptional initiation, particularly in the absence of P-TEFb (positive transcription elongation factor b) [21]. Promoter proximal pausing of transcription is a universal process for RNA polymerase II transcripts that allows modifications of nascent transcripts with 5′ m^7^G caps, proper transcript processing and elongation checkpoint control. In an active transcription process, the cyclin dependent kinase 9 (Cdk9) in P-TEFb may phosphorylate the carboxyl domain of the RNAPII, DSIF and NFLF-E at sites next to its RNA recognition motif, and release the paused transcripts for processive elongation [22] (Figure 2).

NELF consists of four subunits, A, B, C or D, and E. The C and D subunits of NELF are alternatively spliced products of the same gene [23]. Remarkably, the RD protein is identified as the E-subunit or the smallest subunit of the NELF complex [21]. NELF-E is important for NELF function because it has a functional RNA recognition and binding (RRM) domain. Mutations of the RRM domain in NELF-E impair transcription repression without affecting protein-protein interactions with other NELF subunits [23].

The biochemical activities for various NELF subunits were determined mainly by Hiroshi Handa’s team in Yokohama, Japan. They reconstituted NELF-like complex with epitope-tagged NELF-A, NELF-B, NELF-D, and NELF-E expressed in insect Sf9 cells and investigated their interactions [24]. It was shown that NELF binds to DSIF-RNA Polymerase II (RNAPII) complex [21,23]. The leucine zipper of NELF-E interacts directly with NELF-B but not with NELF-A or NELF-D. NELF-D interacts directly with NELF-A and NELF-B. However, NELF-B does not interact directly with NELF-A. Consistent with the model that NELF-B and NELF-D (or NELF-C) that bring NELF-A and NELF-E together via 3 protein-protein interactions, and NELF-A binds to RNAPII directly. The NELF complex needs interactions of 4 subunits, A–(D or C)–B–E. NELF-C and NELF-D are present in distinct NELF complexes [24].

In the nucleus, NELF interacts with the nuclear cap-binding complex (CBC) and participates in the 3′ end processing of mRNA. Both cap-binding proteins CBP-80 and CBP-20 of CBC bind to NELF-E. The region from amino acids 244 to 380 of NELF-E containing an RNA-recognition motif is necessary and sufficient for the interaction between NELF and CBC-RNA. Knockdowns of NELF-E and CBP80 in HeLa cells rendered the free NELF subunits and CBP20 unstable, which are subjected to degradation rapidly, and abolished functions of the NELF holocomplex and the CBC [25].

On the transcription and processing of small nuclear RNA (snRNAs) that are essential for the assembly and functions of spliceosomes, and small nucleolar RNAs (snoRNAs) that are integral to the biosynthesis of ribosomes by RNAPII, NELF-E and ARS2 (arsenite resistance protein 2) play their respective roles progressively through binding to the cap-binding complex CBC. The binding of NELF-E allows promoter proximal pausing, and the subsequent replacement of NELF-E by ARS2 to the same binding site on CBC plus concurrent binding by PHAX (phosphorylated adaptor for RNA export) enable a continuous process to produce these distinct groups of RNA and prepare them for further processing in the cytoplasm or the nucleus [26].

The mRNAs for histones are special as they may not contain polyadenylated 3′ ends. The RRM of NELF-E may bind to the stem-loop structure in the 3′ end of the “replication-dependent” histone mRNAs and enhance their stability. Notably, NELF appears to physically associate with histone gene loci and forms the so-called “NELF-bodies”, which are probably engaged in the processing of histone mRNA. Knockdowns of NELF-E and CBP-80 in HeLa cells led to increased accumulation of the aberrant, polyadenylated form of histone mRNAs [25].

NELF-E was studied on the regulation of gene expression for human immunodeficiency virus HIV. Recombinant protein with the RNA-recognition motif (RRM) of NELF-E was shown to bind the stem of viral RNA with TAR (transactivation response element) [27]. Such binding (between NELF-E and TAR) inhibits the human immunodeficiency virus type 1 basal transcription from the long terminal repeat (HIV-LTR) [28]. Cyclin-dependent kinase 9 (Cdk9) in P-TEFb can phosphorylate NELF-E at sites next to its RNA recognition motif, and thereby making NELF-E no longer able to bind to TAR nor to repress HIV transcription [22].

Poly-(ADP-ribose) polymerase 1 (PARP1) can covalently transfer of ADP-ribose from NAD^+^ onto substrate proteins, consequently modulates diverse biological processes of ribosylated proteins. In a human cell line, Gibson et al. reported that PARP1 interacted with NELF and mediated ADP ribosylation of NELF-E [29,30]. In a manner similar to Cdk9 of P-TEFb-mediated phosphorylation, ADP-ribosylation near the RRM of NELF-E ablates its ability to bind RNA, and therefore releases RNAPII from NELF-dependent pausing. ADP-ribosylation of NELF-E is dependent on phosphorylation by Cdk9/P-TEFb. PARP1-dependent ADP-ribosylation of NELF-E reinforces P-TEFb-mediated RNAPII pause release. Depletion or inhibition of PARP1, or mutagenesis of the four negatively charged glutamate residues in NELF-E (E122, E151, E172, and E374) to glutamines (Q), resulted in a substantial reduction in NELF-E modification by PARP-1. Thus, it was concluded that PARP1-dependent ADP ribosylation of phosphorylated NELF-E (and NELF-A) is necessary for efficient release of RNA polymerase II into productive elongation [29,30].

In a study of double-stranded (ds) DNA break and DNA repair, NELF-E is found preferentially recruited to nearby transcriptionally active genes at sites of DNA ds-breaks to underpin transcriptional repression in a PARP-1 dependent manner. The transcriptional pause allows the machinery for DNA repair to carry out its function and avoid a potential collision of the two systems (DNA repair and transcription) at the damage site. A slight decrease in ribosylation of NELF-E (by PARP-1) at the site of DNA breaks also ensures the pause of transcription imposed by NELF [31].

In a different study of UV-light induced DNA damage (but not ds-breaks) and changes of RNA metabolism, it was found that phosphorylation of NELF-E at serine 115 by MK2 (MAPK activated protein kinase II) promotes the recruitment of 14-3-3 dimers and rapid dissociation of the NELF complex from chromatin. This process is accompanied by elongation of transcription through RNA polymerase II [32].

A cryo-electron microscopy structure with paused transcription complex consisting of RNAPII purified from porcine, human DSIF expressed in bacteria, and human NELF complex expressed in insect cells has been presented with 3.2 Å resolution (Figure 2A). This paused RNAPII–DSIF–NELF elongation complex structure contains a tilted DNA–RNA hybrid that is incompatible with binding of nucleoside triphosphate substrates. The NELF complex binds to the funnel structure of the RNAPII and restricts its mobility. During this paused state, through its two tentacle-like structures, NELF has flexible contacts with the DSIF and the exiting RNA. NELF also prevents the binding of the anti-pausing transcription elongation factor IIS [33].

The same research team continued to show cryo-EM structures of the elongation complex with an active transcription process. To re-activate the paused complex, P-TEFb phosphorylates RNAPII, DSIF and NELF. NELF is displaced by Pol II associated factor PAF. Homolog of suppressor for transposon Ty insertion 6 (SPT6) binds to RNAPII and then set off the transcriptional elongation process (Figure 2B) [34].

Considering the fundamental relevance of NELF-E on the cellular process such as gene expression, dysregulated production or aberrant function of NELF-E could have detrimental effects on life processes. Reports on disease associations of NELF-E are relatively scarce but there are recent articles related to NELF-E in carcinogenesis. NELF-E was preferentially overexpressed in hepatocellular carcinoma (HCC), especially in HCC with portal vein invasion (PVI) [35]. NELF-E via somatic copy number alterations appeared to enhance oncogene MYC signaling and activation, which promoted HCC progression [36]. NELF-E protein levels could be an independent risk factor for early intra-hepatic recurrence of HCC within two years of surgery [35].

### A Gene Within a Gene: miR-1236 in Intron 3 of NELF-E

While overexpression of NELF-E may be associated with carcinogenesis, a genetic element embedded in the NELF-E gene serves as a tumor suppressor. Between nucleotides 2104 and 2205 of the human NELF-E gene is the precursor for a micro-RNA, miR-1236. This is a splice-dependent mirtronic miRNA that co-expresses with NELF-E [37,38]. miR-1236 is transcribed by RNA polymerase II. The promoter region of miR-1236 is the same as the NELF-E promoter.

miR-1236 can inhibit tumors migration, proliferation and invasion activity by binding to target mRNAs and down-regulating their protein synthesis. It can also induce target gene expression by binding to gene promoter in the nucleus. Binding of miR-1236 to target mRNAs can lead to abolition of hypoxia-induced epithelial–mesenchymal transition (EMT) and inhibition of migration and invasion of tumor cells. miR-1236-3p can suppress ovarian cancer metastasis by binding to the 3’ untranslated region of zinc-finger E-box binding homeobox 1 (ZEB1) mRNA, and downregulate ZEB1 gene expression [39]. ZEB1 is an EMT inducers. miR-1236 can inhibit the invasion and metastasis of gastric cancer possibly through decreasing phosphorylation by Ak transforming (AKT) kinase and binding to metastasis-associated protein MTA2 [40]. miR-1236 may bind to the promoter of p21 and induces nuclear p21 expression. p21 is a cyclin dependent kinase inhibitor and a regulator of cell cycle progression. miR-1236 and p21 expression are significantly decreased in renal cell carcinoma. Targeted activation of p21 by miR-1236 can inhibit cell proliferation, may have therapeutic potential in the treatment of renal cell carcinoma [41].

## 3. SKIV2L or SKI2W

### Quality Controls I—Clearing Broken, Bad, Used and Viral RNA Products in the Cell Requires An Engine to Unwind Helical structures of RNA for Degradation—The Helicase for Cytoplasmic Exosomes SKIV2L (Ski2w or DDX13)

The Ski complex is a heterotetrameric complex consisting of Ski2, Ski3 and Ski8 in 1:1:2 stoichiometry that is associated with the cytoplasmic RNA exosome for 3′→5′ mRNA decay [42]. Early investigations revealed that yeast Ski2p is engaged in the inhibition of translation from polyadenylation-negative RNA, and has antiviral activities as yeast viruses are RNA structures without poly-A tails [43]. Mutations in these Ski genes in yeasts gave rise to the super-killing phenotype with viral infections [44,45].

Exosomes are highly conserved complex structures responsible for mRNA turnover and therefore fundamental for regulation of gene expression and cell differentiation in eukaryotes. Each exosome consists of 9 or 10 structural proteins that forms a channel structure by which mRNA molecules thread through and get degraded 3′→5′ by an exoribonuclease [46]. The Ski complex serves to unwind the secondary or helical structures of single stranded mRNA or double stranded RNA from some viruses. The Ski3 and Ski8 proteins form the scaffold structure for the Ski complex by which the Ski2 RNA helicase performs its unwinding activity to prepare RNA to thread through the exosome for degradation in the cytoplasm [47,48]. In the nucleus, a similar exosome structure exists but the corresponding RNA helicase is the Mtr4 or KIAA0052 [49,50].

The functional human SKI2W gene in the HLA was discovered in 1995 [13]. It also documented the existence of a SKI2W-related gene, KIAA0052, in the human genome. Because of the striking sequence similarities with the yeast antiviral gene SKI2, SKI2W was renamed to SKIV2L, and KIAA0052 (MTR4) was renamed to SKIV2L2. The human SKIV2L gene is located between RD (NELF-E) and DOM3Z (DXO) at the MHC class III region. It spans 11 kb and consists of 28 exons that codes for a 4 kb transcript. SKIV2L transcripts are ubiquitously expressed in various tissues, especially in the spleen, lymph nodes, duodenum, appendix and gall bladder. The intergenic region between NELF-E and SKIV2L is close to minimum and it was shown that a 176 bp DNA fragment from this region had bidirectional promoter activity for both NELF-E and SKIV2L [51].

The Skiv2l protein has a molecular weight ~140 kDa that consists of 1,246 amino acids [13,17,52]. The Skiv2l protein has structural motifs characteristic of an RNA helicase, which include motifs for binding and hydrolysis of nucleotide triphosphates such as ATP, RNA binding and a DEVH-box [13,53]. Thus, another name for Skiv2l is DDX13 [54]. Skiv2l uses ATP as an energy source to unwind helical or secondary structures of RNA molecules. In addition, Skiv2l contains a leucine zipper motif for protein-protein interaction [13,54], an RGD motif that may be a ligand for adhesion molecules, and an acidic motif with KDLL sequence for endoplasmic reticulum targeting (possibly for unfold protein response). The Skiv2l protein also consists of four potential N-linked glycosylation sites and two potential sulfation sites [13,17].

Fusion proteins with human Ski2w immobilized on polyvinylidene fluoride (PVDF) membrane were able to hydrolyze γ-^32^P-labeled ATP, which would be an energy source for helicase activities [13]. Immunocytochemistry experiments using specific rabbit antiserum showed that Skiv2l proteins in HeLa cells are localized to the nucleolus and the cytoplasm, a phenomenon that was independently confirmed in transfectants expressing tagged-Skiv2l protein (Figure 3) [52]. Co-sedimentation experiments for ribosome profile through sucrose gradient centrifugation and fractionations revealed that Ski2w is associated with the 40S subunit of ribosomes [52].

The roles of human Skiv2l on the fine regulation of antiviral defense and RNA turnover was examined recently [55]. Discrimination of foreign RNA such as viral RNA with 5′ triphosphate instead of a m^7^G cap for host mRNA, or double-stranded RNA that seldom exists in a eukaryotic cell, are achieved by RNA helicases such as retinoic acid activating gene 1 (RIG1) or melanoma differentiation-associated protein 5 (MDA5) [56,57]. Stimulation with RIG1/MDA5 ligands [such as thapsigargin or poly(I:C)] on primary macrophages with *knockdown* of Skiv2l, but not Xrn1 or Ski3, led to substantial increase of transcription for interferon-β. Thus, Skiv2l specifically suppressed or regulated the production of type I interferon from macrophages in response to ligands that mimic viral infections. This is because Xrn1 knockdown or Ski3 (TTC37) knockdown in similar experiments had no effects on IFN-β expression [56]. Skiv2l probably fine-tunes the antiviral response by RIG1/MDA5 to prevent the over inflammation that can cause tissue injuries.

During a viral infection or an inflammation, the host cells are under stress and endoplasmic reticula in cells are subjected to excessive protein synthesis that overwhelms the posttranslational machinery to properly fold and modify protein molecules. Such stress triggers the unfolded protein response (UPR) through inositol-requiring enzyme 1 (IRE-1) for endonuclease cleavage of specific cellular mRNAs. One of them is a premature mRNA, which retains an unspliced intron, for the *X*-box *b*inding *p*rotein 1 (XBP-1). Splicing out the intronic sequences from the premature mRNA of XBP-1 in the cytoplasm generates ligands for endogenous retinoic acid-inducible gene-like receptors (RLRs). In the absence of Skiv2l, such UPR would trigger elevated production of type I interferon (IFN) production. Therefore, SKIV2L is an important negative regulator of the RLR response to exogenous RNA ligands including RNA viruses [56]. Under such a scenario, the absence or partial functional deficiency of Skiv2l could generate unchecked or dysregulated production of ligands or sensors for RLR. The results would be autoinflammation in the absence of a viral infection, or increased susceptibility to systemic autoimmune disease. Table 2 lists results of recent attempts to investigate the potential roles of Skiv2l mutations in human diseases.

It was mind-boggling to find that a deficiency in Skiv2l or another Ski complex protein Ski3 or TTC37 causes the syndromic intractable diarrhea known as tricho-hepato-enteric syndrome or THES [58,59]. This extremely rare syndrome is characterized by (1) the presence of woolly and easily breakable scalp hair, (2) facial dysmorphism and hypertelorism of paired organs such as eyes, (3) intrauterine restriction in growth with low birth weights, (4) dysfunctional liver with hematochromatosis and coarse appearance, (5) intractable diarrhea that may initiate soon after birth or at neonatal stage, persists despite prolonged bowl rest. Many baby patients with THES require parenteral nutrition, become dehydrated and marasmic, and incapable to thrive [60,61,62,63]. 

Further analyses revealed that most THES patients had defects in the immune system with low levels of switched immunoglobulins, highly variable responses to vaccination including unresponsiveness of specific antigens, impaired T cell and NK cell functions such as low production of γ-interferon and abnormal T cell proliferation, and severe Epstein Barr Virus (EBV) infection [77]. It appears that 60% of the THES patients had a genetic deficiency of TTC37 [59] and 40% had a deficiency of SKIV2L [78], and many of those homozygous deficiencies were the results of consanguineous marriage. Since the discovery of THES, a spectrum of disease phenotypes and severity has emerged, and the disease has been reported in multiple racial groups [74,75,79,80,81]. It has been noticed that THES patients with deficiencies of SKIV2L experienced more severe disease in terms of liver damage and impairment of prenatal growth than those with deficiencies of TTC37 [78]. THES patients with deficiency of Skiv2l but not deficiency of TTC37 exhibited a remarkable type I interferon stimulated gene expression profile [56].

It is not clear for the mechanism between the loss of function for SKIV2L or TTC37 and the detrimental phenotypes of THES, which include the failure to thrive, inability in food digestion, poor quality in hair texture, and impairments of liver and immune functions.

Intriguingly, elevated expression levels of type I interferon stimulated genes in the absence of apparent viral infections is a marked phenomenon of systemic autoimmune diseases including systemic lupus erythematosus and juvenile dermatomyositis [82,83,84]. Thus, it would be appropriate to examine if there were genetic or acquired deficiency, dysregulation or impaired functions of Skiv2l in autoimmune, inflammatory and infectious diseases [56].

## 4. DXO or DOM3Z

### Quality Controls II—Destroy Disqualified Products—DXO (DOM3Z) to Degrade Miscapped or Uncapped RNA from the Start to the End

The discovery of human DOM3Z (DXO) was reported by Yang and colleagues in 1998 [14]. This gene is arranged head-to-head orientation with STK19 (RP1) and tail-to-tail with SKIV2L (SKI2W) [15,17]. The 5′ ends of DOM3Z and STK19 overlap but they are in opposite configurations. The exon 1 of STK19 is present at the intron 2 of DOM3Z genomic region. The 3′ ends of DOM3Z and SKI2W are separated by 59 base pairs.

The DOM3Z gene has CpG-rich sequences at its 5′ region, a feature conserved in housekeeping genes in simple and complex eukaryotes. Human DOM3Z is ubiquitously expressed, but the highest expression levels are found in the adrenals and the testes. The first exon (123 bp) of DOM3Z is noncoding because there is an in-frame TGA stop codon at nucleotides 19–21. The full length DOM3Z cDNA is 1386 bp and its genomic DNA is 2.2 kb in size, which consists of 7 exons encoding 396 amino acids. Between residues 103 to 124 of the Dom3z protein is a leucine zipper motif for protein-protein interactions [14]. Yang and colleagues also found that human DOM3Z has homologs in round worm *Caenorhabditis elegans* (DOM-3), baker’s yeast, fission yeast and possibly flowering plant Arabidopsis [14,15,17]. However, no other known protein motifs were identified at that time.

Most progress in studying the biologic functions of DOM3Z homologs occurred in yeasts, which is named **Rai1**. Through genetic deletion and complementation analyses, Xue and coworkers showed in year 2,000 that Rai1 interacts Rat1 (also named Xrn2). The yeast Rat1 is a nuclear 5′→3′ exoribonuclease that is important in RNA processing and RNA degradation [85]. Rai1 stabilizes and activates the exonuclease activities of Rat1. Genetic mutants of Rai1 affects the proper (5′ and 3′) processing of the 5.8S ribosomal RNA. Deletion of Rai1 gene leads to defective biogenesis of the 60S ribosome but that can be suppressed by high copy of RAT1. Intriguingly, double mutants of Rai1 and putative RNA helicase gene Ski2 in yeast were synthetically lethal [85], suggesting Rai1 and Ski2 proteins have overlapping functions that cannot be both dysfunctional.

Xiang, Jiao, Tong, Kiledjian and colleagues in 2009 reported crystal structures of *S. pombe* Rat1 in complex with Rai1 in 2.2 Å resolution [86], and Rai1 and its murine homolog Dom3z in 2.0 Å resolution. It was shown that Rai1 can stabilize Rat1 secondary structure to facilitate Rat1 to degrade RNAs more effectively. They identified a large pocket in Rai1 and in Dom3z that contains highly conserved residues, including three acidic side chains that coordinates a divalent cation such as Mg^+2^ or Mn^+2^. They demonstrated that Rai1 has pyrophosphohydrolase activity towards 5′ triphosphorylated RNA [86]. This team of investigators continued to demonstrate that Rai1 possesses decapping and endonuclease activities, which can remove the entire cap structure dinucleotide from an mRNA in vitro and in yeast cells, especially towards mRNAs with unmethylated, faulty or aberrant caps [87]. Thus, it is proposed that Rai1 is involved in quality control process to ensure mRNA 5′-end integrity. The same research team also found a related protein for Rai1 in yeast known as Ydr370C, which is mainly expressed in the cytoplasm. The crystal structure of the Ydr370C protein from *Kluyveromyces lactis* (named as DXO1) was solved and its biochemical activity investigated [88]. Ydr370C has robust decapping activity for incompletely capped mRNA and 5′→3′ distributive exoribonuclease activity, but no activity for pyrophosphohydrolase. The cytoplasmic Ydr370C was renamed Dxo1 because of its decapping and exoribonuclease activities for mRNA. If both Dxo1 and Rai1 in yeast are disrupted, mRNAs with incomplete caps are produced and readily detectable under normal growth conditions. Therefore, yeast possesses two partially redundant proteins that detect and degrade incompletely capped mRNAs [88]. In mammals including humans, there is only one homolog for Rai1 and Dxo1, Dom3z. Dom3z possesses all three functional activities of Rai1 and Dxo1, which are pyrophosphohydrolase, decapping, and 5′ to 3′ exoribonuclease activities. Thus, Dom3z is given an alternative name as Dxo. Dom3z preferentially degrades defectively capped and inefficiently spliced pre-mRNAs (Figure 4A) [89,90]. Collectively, Rai1, Dxo1 and Dxo identify a novel class of proteins that functions in a quality control mechanism to ensure proper mRNA 5′-end fidelity or integrity [89].

If the first (^m7^GpppN_m_pRNA) or the second nucleotide (^m7^GpppN_m_pN_m_pRNA) of the mRNA were methylated at the ribose 2′-O position, it would be protected from Dxo-mediated decapping and degradation [92].

Over the past forty years, it has been firmly established that mRNA in eukaryotes are predominantly modified by a 7-methyl-guanosine cap at the 5′ end during the initiation of gene transcription. This m^7^G cap helps the processes for (a) RNA splicing to remove intronic sequences, (b) the nucleus-cytoplasm transport of mature mRNA for translation, and (c) protection of mature mRNA from degradation. It has been presumed that such capping process in eukaryotes was constitutive and efficient with little errors until yeast Rai1 is shown to play an important role on the quality control of mRNA capping and stability. Moreover, it has been found recently that some bacteria RNA can be capped at the 5′ end with nicotinamide adenine dinucleotide (NAD^+^) to increase their stability. Unexpectedly, such NAD^+^ caps are also present in 5–10% eukaryotic mRNA and some small nucleolar RNA (snoRNA). Notably, NAD^+^-capped mRNAs are *not* translatable. They are also less stable and subject to Dxo-mediated decapping and 5′ →3′ decay, a process known as deNADding (Figure 4B) [91,93].

## 5. STK19 (RP1/G11)

### An On-and-Off Switch—The Nuclear Kinase STK19 (RP1 or G11) Whose Malfunction Can Cause Melanoma

While the 5′ regulatory regions of the duplicated C4 genes were characterized, a canonical polyadenylation signal (AATAAA) was found in the promoter region of each C4 gene, suggesting the presence of a novel gene. Thus, cDNA clones were isolated and genomic DNA sequences for this novel gene were determined [10,11,94]. This gene was originally named RP to memorialize the late Professor Rodney Porter, in his research unit the human MHC complement genes were cloned and characterized.

In human, an intact RP1 gene is located 611 base pairs upstream of the first C4 gene, and a partially duplicated RP2 gene is present upstream of each of the succeeding C4 gene (Figure 1) [10,17]. Immediately upstream of RP1 is DOM3Z. RP1 is in head-to-head configuration with DOM3Z, and tail-to-head with C4 [17]. Upstream of RP2 is another partially duplicated gene fragment for tenascin X, TNXA. Human RP is part of a four-gene duplication complex in the HLA termed the RCCX module, which stands for RP, C4, CYP21 and TNX (Figure 1) [95,96].

RP1 is expressed in all tissues and its highest expression is in the adrenals and the testes of human and rodents. Homologs of RP1 are predominantly present among vertebrates but an exception is found in a sea anemone (*Exaiptasia pallida, LOC110250916/*XP_020913230.1). The RP1 protein contains a bipartite nuclear localization signal and was shown to be a nuclear protein [10]. Subsequently, recombinant protein of this gene, which was also named G11 by Duncan Campbell and colleagues [97], was shown to be a serine/threonine kinase [98]. This gene was renamed to STK19, which uncoupled publications for some of the original observations [10,95,99]. Other names shown at the NCBI website for this gene include D6S60; D6S60E; HLA-RP1.

STK19 gene contains 7 or 9 exons. The 9-exon STK19 gene is 11 kb in size and codes for 364 amino acids with a calculated molecular weight of 41.5 kDa. This long form of STK19 gene overlaps to the 5′ region of the DOM3Z gene. There is a variant coding for 368 amino acids that is due to an alternative splicing, which results in an extension of 12 nucleotides from the 3′ end of exon 5 to the intron 6, leading to the addition of four more amino acids after R221 (VCDC). The 7-exon version of STK19 is 9.1 kb in size that has a 267 bp intergenic distance from DOM3Z. It codes for 254 amino acids with a calculated molecular weight of 30 kDa. The first 110 amino acid residues encoded by the first two exons are absent in the shorter RP1 transcript. Murine STK19 has the small version with 254 amino acids.

Scattered in the introns of the human RP1 gene are 8 copies of short interspersed nuclear elements named *Alu*. One of them is an integral component of a mobile retroelement SVA that consists of a distinct SINE, 21 copies of GC-rich variable number tandem repeats (VNTRs), and an Alu element [10,15,17]. The SVA element in RP1 was flanked by short direct repeating sequences of 13 nucleotides, which was indicative of insertion by a discrete composite genetic unit (Figure 1B). The SVA in RP1 was the first identification and documentation of such distinct and discrete composite retroelements into a human genome [10]. The highly repetitive nature of the SVA sequences render specific biologic studies challenging. It was demonstrated certain member(s) of SVA has the capability to “retrotranspose” with help of the reverse transcriptase coded by active long interspersed element LINE-1 [100,101,102].

In terms of gene function, STK19 encodes a nuclear protein [10] that is a kind of serine/threonine kinase. The STK19 protein can bind ATP at its kinase domain, phosphorylate certain proteins at Ser/Thr residues, such as α-casein and histones [98]. The actual physiological properties and function, the substrates of the kinase activity for STK19, and the relationship with disease for STK19 are being gradually revealed [103]. Results of the interactome analyses revealed that nine proteins bind to STK19 (https://www.ncbi.nlm.nih.gov/gene/8859, [104]). These proteins are DNA polymerase subunit E or POLE, RNA polymerase II subunit G or POLR2G, transcription factor SP3, mRNA splicing factor SF3BF4, basic helix-loop-helix protein e40 or BHLHE40 that controls circadian rhythm and cell differentiation, inosine monophosphate dehydrogenase 1 or IMPDH1 that is an enzyme essential for guanine nucleotide biosynthesis through catalysis of synthesis for xanthine monophosphate from inosine 5-monophosphate, membrane transporter for nucleosides or solute carrier 29 or SLC29A1, preferentially expressed antigen in melanoma PRAME that acts as a repressor of retinoic acid receptor and may promote growth of cancer cells, and tripartite motif protein TRIM23 that may be a ADP ribosylation factor for guanine nucleotide binding protein and plays a role in the formation of intracellular transport vesicles and for their movement from one compartment to another in the cell. These interacting proteins may be substrates or cofactors of STK19 kinase, implicating that STK19 could be engaged in DNA replication, RNA transcription, RNA splicing, nucleoside/nucleotide synthesis and transport, and control of gene expression for development and differential cellular functions.

While STK19 knockdown in cells revealed little effects on transcription, it is required for recovery of transcription after DNA damage and therefore important for the transcription-related DNA damage response [105]. STK19 interacts with Cockayne syndrome B protein (CSB, also named ERCC6) after DNA damage. Cells lacking STK19 were UV-sensitive.

By analysis of large-scale exome data from melanoma and uninvolved tissues, Eran Hodis and colleagues showed STK19 is a novel melanoma gene for its somatic hotspot of point mutations. While the D89N mutation of SKT19 has a 5% frequency in a melanoma [106], no SKT19 mutations were detectable in nevus associated-melanomas [107]. There are 10% gene mutations in STK19 in basal cell carcinoma (BCC). Strikingly, all of those mutations resulted in D89N substitution [108]. Recently, Yin and colleagues reported STK19 can phosphorylate NRAS at Ser-89 and activate oncogenic NRAS to promote melanoma genesis. When STK19 has the D89N change, it is a gain of function mutation that phosphorylates or interacts more efficiently with NRAS and enhances melanocyte transformation [109].

In addition, by next-generation sequencing and bioinformatics analysis, STK19 was found to have novel T/C missense single-nucleotide variants in Chr6:31947203 in induced pluripotent stem cell (iPSC) lines, which can be used to trace the genotype of original cells [110].

Through meta-analyses of published GWAS data, STK19, SKIV2L and NELF-E appeared to be potential pleiotropic genes for metabolic syndrome and inflammation [111]. Moreover, Amare et al. reported that genetic variant of STK19 gene correlated with schizophrenia by bivariate GWAS [112]. Other functions of STK19 remain to be demonstrated.

## 6. The Physical Linkage of NELF-E and SKIV2L (RD and SKI2W), DXO and STK19 (DOM3Z and RP1) and the Emergence of NSDK Quartet in the MHC

We ask when did the genes for NELF-E, SKIV2L, DXO and STK19 start to link together physically as a group in the animal kingdom, and their relationships with the class I and class II genes in the emergence of the MHC in the vertebrates. We attempted to obtain answers to these questions through analyzing data deposited in the National Center for Biotechnology Information (NCBI/PubMed: https://www.ncbi.nlm.nih.gov/home/ or https://www.ncbi.nlm.nih.gov/pubmed/).

The evolution history of NELF-E is intriguing because the presence of a leucine zipper and the RRM, which is a characteristic of a NELF-E like molecule, can be found in *Drosophila* but there are no RD repeats (https://www.ncbi.nlm.nih.gov/protein/NP_648241.1, [113]). The presence of the RD (Arg-Asp) and/or RE (Arg-Glu) repeats is definitive in zebrafish NELF-E (https://www.ncbi.nlm.nih.gov/protein/XP_005170084.1, [114]). Related gene or protein function for SKIV2L can be found in yeast, and for DXO in *C. elegans* [115]. STK19 homologs with high sequence identities are predominantly present among vertebrates. There is no evidence for a physical linkage for any two of the NSDK quartet prior to vertebrates.

For the data deposited in the NCBI, most chromosomal segments are relatively short, and their linkage status is not yet known. The neighboring gene linkages of NELF-E and SKIV2L in one chromosomal segment, and DXO and STK19 in another chromosomal segment are observable in shark, zebrafish, and coelacanth. Genes in each of these pairs are organized in “head-to-head” configuration and therefore they share the same 5′ regulatory sequences in opposite orientations (Figure 1B). The physical linkage of all four genes (i.e., NELF-E, SKIV2L, DXO and STK19) together as a group seems to be definitive in frogs [116]. Again, we should emphasize that data available for analyses are limited.

In egg-laying mammal platypus, MHC class I and class II genes are present on chromosome X3, but most (if not all) of the class III genes form a linkage unit on chromosome X5. The assembly of the class III genes (including NSDK with complement convertases), class I genes and class II genes as the MHC locus appears intact in placental mammals.

While the NSDK quartet genes express ubiquitously, it is not known whether there are concerted controls of gene expression (such as genes in operons in prokaryotes), or differential responses to regulation during growth and development.

## 7. Conclusions

NELF-E-SKIV2L-DXO-STK19 is a complex genetic unit located between complement factor B and component C4 in the class III region of the major histocompatibility complex (MHC) [98,117]. NELF-E, SKIV2L and DXO make important roles in transcription, processing and degradation of a variety of RNAs and antiviral immunity [24,56]. As such, NELF-E or RD is a subunit of the negative elongation factor (NELF). It is an RNA-binding protein that imposes a promoter proximal pause in transcription by the RNA polymerase II. The purpose is to allow capping and quality checking prior to full strength elongation of RNA for gene expression [24,30]. SKI2W or SKIV2L RNA helicase, an engine to unwind helical structures of RNA molecules and propel them through the tunnel formed by cytoplasmic exosomes for 3′→5′ degradation. SKIV2L is an important negative regulator of the RIG-I-like receptors (RLR) mediated antiviral response and an important modulator of the type I interferon response in the cytoplasm [56]. The RLR [56,57], toll-like receptors, and complement systems are innate immune response effectors that engage in pattern recognitions of differentially modified biomolecules to differentiate self and non-self [118,119].

Genetic studies revealed that homozygous deficiency of human SKIV2L contributes to the pathogenesis of syndromic diarrhea or trichohepatoenteric syndrome (THES). Regulation of gene expression is critical in determining cell identity, development and responses to the environment. DXO or Dom3z possesses pyrophosphohydrolase, decapping and 5′-3′ exoribonuclease activities that can preferentially degrade incompletely 7-methylguanosine (m^7^G) capped, or NAD^+^ capped mRNAs from the 5′ end [89]. STK19 is a Serine/Threonine nuclear protein kinase. STK19 is important for the transcription-related DNA damage response [105,109]. STK19 phosphorylates N-RAS at Ser-89. Dysregulated phosphorylation of NRAS at S89 activates and promotes melanoma genesis [109]. It is necessary to point out that the biochemical and functional properties of STK19 are still largely unexplored. It is also not clear whether and how do the NSDK quartet proteins interact with each other, and the advantage for these four genes being tightly linked together.

Thus, NSDK seems to engage in the surveillance of host RNA integrity at the “head” and at the “tail” [120], in the destruction and turnover of faulty or expired RNA molecules or RNA viruses, and in the fine-tuning of innate immunity. The roles of NSDK in carcinogenesis, infectious and autoimmune diseases are only starting to emerge. We look forward to further studies on disease associations, functional mechanisms and therapeutics of these very old but non-classical MHC genes.

## Figures and Tables

**Figure 1 cells-08-01008-f001:**
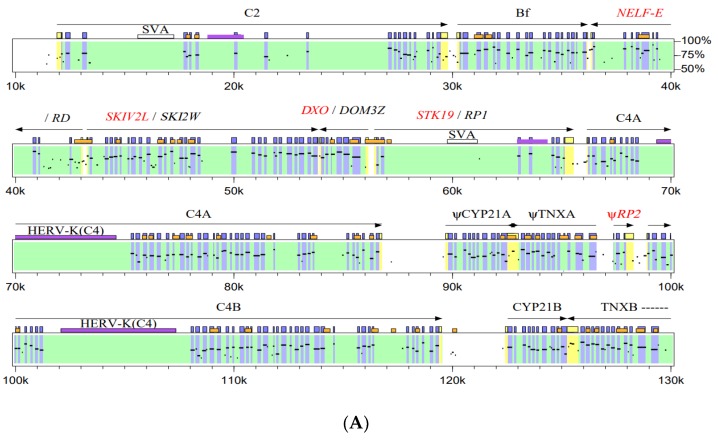
Map and gene structures of the NSDK (NELF-E, SKIV2L, DOX and STK19) quartet in the major histocompatibility complex (MHC) complement gene cluster. (**A**) A molecular map comparing the gene organization of human and mouse MHC complement gene cluster with RNA surveillance quartet consisting of NELF-E, SKIV2L, DXO and STK19 at the intergenic region between complement factor B and complement C4A gene. Original gene names are shown as alternatives. The mouse sequences with identities from 50–100% are plotted below the human sequences. Exons for human genes are shown as solid, purple boxes. The genomic region spanned for each gene is shaded green. The 5’ and 3’ untranslated sequences are in yellow boxes. The conserved coding sequences between human and mouse genes are highlighted in purple. Locations of CpG rich dinucleotides are shown as orange boxes. Large, repetitive retroelements SVA and endogenous retrovirus HERV-K(C4) are shown in burgundy. Arrows with solid lines represent configurations of structural genes. *RP2* and *TNXA* are partially duplicated gene fragments. *CYP21A* is a pseudogene in human. Pseudogenes and gene fragments are labeled with ψ in front of gene names. Genomic DNA sequences are obtained from the following accession numbers: Human, U89335-U89337, AF019413, M59815, M59816, L26260-L26263, U07856, AF059675 and AF077974; mouse, AF030001, AF049850 and AF109906. Numberings below box represents length in human genomic DNA in kb (2k = 2 kb). The 5′ region of complement C2 gene is not well defined. Located approx. 26.4 kb upstream of the gene for C2 isoform 5 and in opposite orientation is the zinc finger and BTB domain containing protein 12 (ZBTB12). Present at the 3′ end of the C2 gene and in opposite orientation is a genetic element for long non-coding RNA (lncRNA C2-AS1, not shown). (**B**) Exon-intron structures of human NELF-E, SKIV2L, DXO and STK19 (modified from refs [15,16]).

**Figure 2 cells-08-01008-f002:**
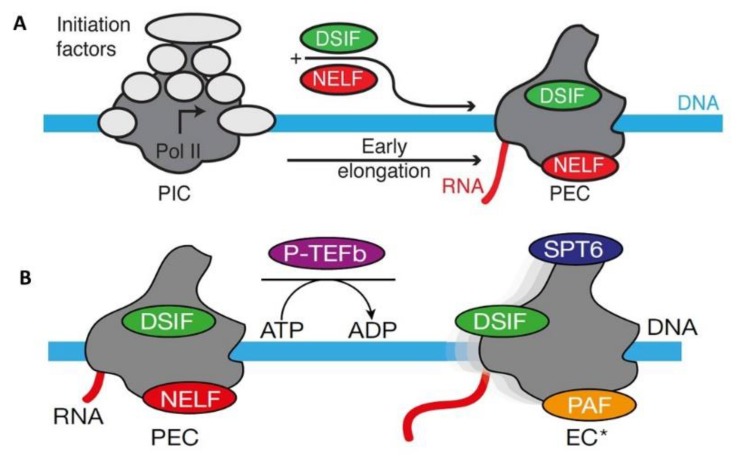
Pause and go of RNA polymerase II during transcription. (**A**) Schematic of conversion of the RNAPII pre-initiation complex (PIC) to a promoter proximally paused RNAPII-DSIF-NELF elongation complex (PEC) [33]. (**B)** Schematic showing conversion of the paused PEC to the activated RNA Pol II-DSIF-PAF-SPT6 elongation complex (EC*). SPT6, homolog of yeast *s*u*p*pressor of *T*y*6* (adapted from references [33,34], with permission from Dr. Patrick Cramer, Max Planck Institute for Biophysical Chemistry, Germany).

**Figure 3 cells-08-01008-f003:**
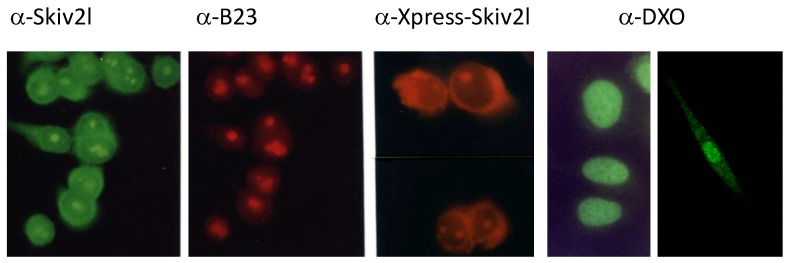
Cellular localization of Skiv2l and DXO in HeLa cells under immunofluorescent microscopy. Human HeLa cells were stained by antibodies against human Skiv2l (Ski2w), nucleoli-specific B23 (nucleophosmin), anti-Xpress for HeLa transfectants with Xpress-Skiv2l fusion protein, and antibodies against DXO (Dom3z). The first three panels illustrate the presence of Skiv2l in the nucleoli and cytoplasm. The fourth and fifth panels show the dominant location of DXO (Dom3z) in the nuclei of HeLa cells. Photographs were originally taken from UV-microscope with 10 × 40 magnifications (modified from [15,52,55]).

**Figure 4 cells-08-01008-f004:**
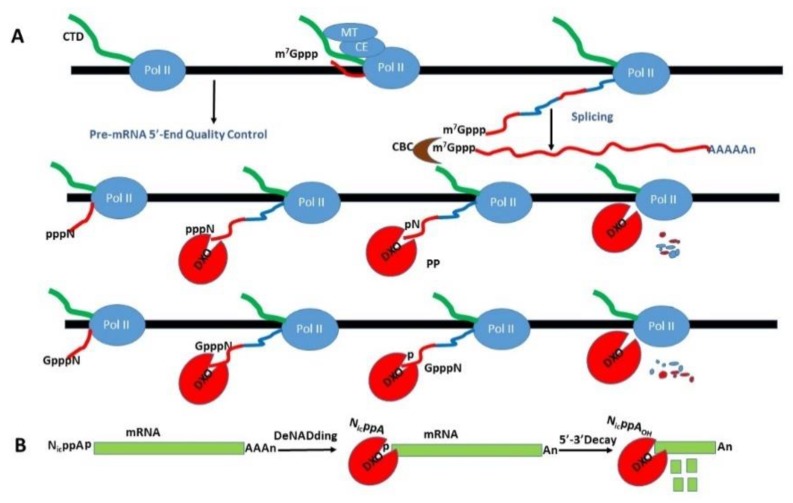
DXO decapping and DXO-mediated decay of NAD^+^-capped mRNA. (**A**) DXO can remove the diphosphate from a triphosphorylated cap of mRNA, the entire cap structure of an unmethylated guanosine-capped mRNA, or aberrant pre-mRNAs. (**B**) DXO hydrolyzes the phosphodiester linkage 3′ to the adenosine to remove the entire NAD^+^ moiety and generates an RNA molecule with 5′ monophosphate. Modified and redrawn from ref [89] for panel A, and ref [91] for panel B (with permission from Dr. Mike Kiledjian, Rutgers University, NJ).

**Table 1 cells-08-01008-t001:** Characteristics of the NSDK genes for RNA metabolism and surveillance quartet in the class III region of the human major histocompatibility complex.

Official Symbol	Original Name; Other Earlier Names	Gene, Transcripts and Proteins (Location, cDNA, aa)	Expression	Specific Features	Function	Website
**NELF-E**	**RD** RDP; RDBP; D6S45	11 exons 6.7 kb 380 aa	Ubiquitous; highest in testis	CpG rich, leucine zipper motif, RD repeats, RNA recognition motif (RRM); miR-1236 in intron 3	A subunit of the negative RNA elongation factor that represses transcriptional elongation by RNA polymerase II	https://www.ncbi.nlm.nih.gov/gene/7936
**SKIV2L**	**SKI2W** HLP, SKI2, 170A, DDX13, DDX49, SKIV2, SKIV2L1, THES2	28 exons 11 kb 1246 aa	Ubiquitous; highest in spleen	CpG rich, leucine zipper motif, helicase domain, RGD and Alu element	Unwinds RNA secondary structures, RNA turnover, antiviral defense, modulates type 1 interferon expression	https://www.ncbi.nlm.nih.gov/gene/6499
**DXO**	**DOM3Z** DOM3L, NG6, RAI1	7 exons 2.2 kb 396 aa	Ubiquitous; highest in testis and adrenal gland	CpG rich; leucine zipper motif	RNA quality control, decapping and 5′→3′ RNA decay	https://www.ncbi.nlm.nih.gov/gene/1797
**STK19**	**RP1** and **RP2** HLA-RP1, G11, D6S60, D6S60E	9 exons 11 kb 364 aa or 7 exons 9.1 kb 254 aa	Ubiquitous; highest in adrenal gland	Partial gene duplication in the RCCX modules; SVA (SINE - 21 CpG rich VNTRs-Alu) in intron; 8 Alu elements	Nuclear Serine/Threonine kinase	https://www.ncbi.nlm.nih.gov/gene/8859

**Table 2 cells-08-01008-t002:** A list of human diseases associated with mutations of SKIV2L.

Disease	SNP	Relationship to Disease	Remarks
SLE (Systemic Lupus Erythematosus)	T allele of rs419788 in intron 6	Confer disease susceptibility in additive pattern	One copy confers a low risk of disease and two copies result in greater susceptibility [64]
AMD (Age-related macular degeneration)	R151Q (rs438999)	May exert a functional effect in AMD	Strong LD (linkage disequilibrium) with Bf R32Q (rs641153) [65]
	Intronic SNP (rs429608)	AMD genetic protective factors	Protective effect for AMD [66] in Han Chinese and Japanese populations [67,68]
	rs429608 and rs453821	Significantly associated with neovascular AMD	Not associated significantly with PCV [69]
PCV (Polypoidal Choroidal Vasculopathy)	3′UTR (rs2075702)	Significant association with PCV	Decreased risk of developing PCV [70]
SD/THES (Syndromic diarrhea/tricho-hepato-enteric syndrome)	c.1635_1636insA (p.Gly546Argfs*35) c.2266C>T (p.Arg756*) c.2442G>A (p.Trp814*) c.848G>A (p.Trp283*) c.1022T>G (p.Val341Gly) c.2572del (p.Val858*) c.2662_2663del (p.Arg888Glyfs*12) c.1434del (p.Ser479Alafs*3)	Deleterious mutations detected in six individuals with typical SD/THES	Molecular defects in SKIV2L cause SD/THES [58,71]
	c.1891G>A p.Gly631Ser c.3187C>T p.Arg1063*	Two new mutations found in a trichohepatoenteric syndrome patient	The patient was a Malaysian child [72]
	(c.1891G>A) (c.1120C>T)	Two variants identified in THES patients	These two mutations can cause THES [73]
	c.1420G>T (p.Q474*) c.3262G>T (p.E1088*)	Novel compound heterozygous nonsense mutations were identified in THES patient	Decreased levels of SKIV2L protein expression in blood mononuclear cells [74]
	25 exons (p.Glu1038 fs*7 (c.3112_3140del))	A rare mutation in THES patient	This patient died at three year old [75]
IBD (Inflammatory Bowel Disease)	c.354+5G>A	Identified a novel splicing mutation in a patient with IBD (ulcerative colitis)	This mutation was related to Inflammatory Bowel Disease [76]

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
