# Peer review of "An RNA Metabolism and Surveillance Quartet in the Major Histocompatibility Complex"

_cells, 2019, doi:10.3390/cells8091008_

Round 1

Reviewer 1 Report

This is a comprehensive and detailed review of the discovery and function of an important cluster of genes in the MHC; and is an appropriate addition to the literature summarising the organisation of this gene-dense region of the genome.

I would suggest that, before publication, the authors try to make the manuscript more accessible to the non-expert reader by:

a) Taking the time to define the many abbreviations.  For example, the definition of NSDK found on line 52 should appear in the abstract where the term first appears; RD should be better defined ("so called because of its 23 consecutive arginine-aspartate repeats"?); RP should be defined (arginine-proline repeats?); define RLR ligands (line 277); and so on.

b) Taking time to explain some of the complex biology more clearly.  Some sections of the review assume a lot of prior background knowledge which the average reader is unlikely to have . One example is the biology of "RLR" mentioned above.  Related to this is the section on type 1 interferon biology: as it is, the link between Skiv21 dysfunction and diarrhea is totally obscure.

Although the majority of the article is well written and uses a semi-formal style (which actually enhances its readability), some of the grammar and vocabulary is incorrect.  One memorable example is on line 206 (abolishment for abolition)

Finally, I wonder if it might be preferable to refer to this cluster of genes as a "region" or "subregion" rather than "quarter". 

Author Response

First reviewer:

Comments and Suggestions for Authors

This is a comprehensive and detailed review of the discovery and function of an important cluster of genes in the MHC; and is an appropriate addition to the literature summarising the organisation of this gene-dense region of the genome.

I would suggest that, before publication, the authors try to make the manuscript more accessible to the non-expert reader by:

a) Taking the time to define the many abbreviations. For example, the definition of NSDK found on line 52 should appear in the abstract where the term first appears; RD should be better defined ("so called because of its 23 consecutive arginine-aspartate repeats"?); RP should be defined (arginine-proline repeats?); define RLR ligands (line 277); and so on. b) Taking time to explain some of the complex biology more clearly. Some sections of the review assume a lot of prior background knowledge which the average reader is unlikely to have. One example is the biology of "RLR" mentioned above. Related to this is the section on type 1 interferon biology: as it is, the link between Skiv21 dysfunction and diarrhea is totally obscure.

Although the majority of the article is well written and uses a semi-formal style (which actually enhances its readability), some of the grammar and vocabulary is incorrect. One memorable example is on line 206 (abolishment for abolition)

Finally, I wonder if it might be preferable to refer to this cluster of genes as a "region" or "subregion" rather than "quarter".

Responses: We highly appreciate the suggestions by the reviewers and have made every effort to improve the readability of the manuscript.

We have added a section after the abstract entitled “Abbreviations” to define all abbreviations used this manuscript. In addition, we define all abbreviated terms in the text at their first usage. We have elaborated description of the disease caused by deficiency of SKIV2L and TTC37, Tricho-Hepato-Enteric Syndrome (lines 333-355) We have slightly revised the description of Skiv2l on antiviral defense and unfold protein response (lines 324-332). On the title of the article, we have revised it to “An RNA Metabolism and Surveillance Quartet in the Major Histocompatibility Complex”- i.e. “quarter” was changed to “quartet”. An English usage error “abolishment” was corrected to “abolition” – line 247.

Reviewer 2 Report

The review entitled ‘An RNA Metabolism and Surveillance Quarter in the Major Histocompatibility Complex‘ updates some of the information about 4 interesting genes in the human MHC class III that were discovered in part by the co-author CY Yu and colleagues in the 1990s. This cluster of 4 adjoining genes NELFE, SKIV2L, DXO, STK19 is referred to as the NSDK multiple gene region. The review is interesting and important because it attempts to connect the regulation of self and nonself (viral infections) transcription and translation with some of the genes of the HLA class III region that are involved in the immunological RNA world. Immunologists when reviewing the overall structure, gene interactions and functions of the human MHC genomic region often neglect the details of this important gene region between C2 and C4.

In general, the review is clearly written and covers most publications and aspects of the NSDK gene region that have emerged during the last 25 years. However, there are a few points in the review that are unclear or have been intentionally or accidently omitted, but that could be reconsidered or added for a clearer and more comprehensive evaluation of the role and interactions of the NSDK genes.

1. It is not clear whether the NSDK gene region is an operon regulated by similar or different promoters and enhancers. Some conclusion or speculation could be made in regard to this possibility. Is the NSDK gene region a convenient designation simply based on their proximity or is it an example of an eukaryotic operon?

2. What are the quotes in italics beneath the headings and where do they come from (Ref?)? Eg., p 2, lines 60-61; p 6, lines 217-219; p 9, lines 293 – 294; p 11. Lines 363; 

3. Figs 1 and 3 are too small; difficult to read the text within figs, eg for Fig 1 Alu, numbers, whatever is written in the helicase domain, etc , impossible to read without magnification. This problem applies to some of the other figures as well.

4. Table 1 needs references or a reference column.

P2, Fig 1A. if possible, please add and label the locations of the important sequences ZBTB12 and C2-AS1 within the C2 gene structure.

P2, line 64. The CFB in the text is Bf in Fig 1. Consistency with terminology required.

P2, line 65. What transcription factors bind to the 5’ regulatory region of the RD gene?

P3, line 92. What types of RNA are known to bind to the RRM (mention here)?

P3, line 98. What chromosome is DRB (DSIF) on?

P4, line 105. Is this NFLF-E or should it be NELF-E? What is the difference if they are different?

P4, line 107. What NELF genes produce the A, B subunits and where are they relative to NELF-E gene? Please indicate where the NELF-C-D gene is located.  Is their expression coordinated? What co-ordinates their expression?

P4, line 111. Delete ‘the’

P4, line 131. What is ARS2? This has suddenly appeared from nowhere with no prior explanation. Same with PHAX? Maybe a Table listing the interacting proteins, genes and gene locations would be an aid to a better understanding of the function(s) of NELF-E.

P5, Fig. 2. Why is EC starred * (EC*) in the text of the figure?

P6, line 193-194. NELF suppression results in overexpression of HepC  virus and this also may be related to hepatocellular carcinoma (Iida M, et al. Oncol Rep, 2012 Aug. PMID 22614758). Also, NELFE might regulate HIV transcription as described in the text and NELFE knockdown increases HIV transcription (Rao JN, et al. Biochem J, 2006 Dec 15. PMID 16898873). 

P6, 198. miR-1236 is a negative regulator of VEGFR-3 signaling (Jones, Li et al 2012 ATVB,  doi: 10.1161/ATVBAHA.111.243576). This is interesting in relation to the treatment of macular degeneration (MD) and cancers with antiVEGF therapy ( PMID 26857947doi:10.1101/cshperspect.a006577)and raises a possible role of miR-1236 in the treatment or regulation of MD.  miR-1236 can reduce hepB virus replication and protein production (Huang et al 2016, doi: 10.1038/srep34740) 

P6, line 210. Grammar. ZEB1 is an … inducer (not inducers).

P8, Table 2. GWAS also have implicated C2/C2-AS1, C3, CFB (doi: 10.1073/pnas.0912702107) and VEGFA (Yu et al 2011,  doi: 10.1093/hmg/ddr270) in age-related macular degeneration (AMD). Are the high risk SNP of these genes in LD? Are there haplotypes linking the AMD SNP of C2/C2-ASI, CFB, NELFE, SKIV2L in the class III region with VEGFA?

P 11, 375. Delete an ‘is’ and express as ‘RP1 is expressed ubiquitously in ..’

P12, 395 – 399.  The two SVAs located in the C2 and the STK19 genes are only about 30 kb apart (Fig. 1). Do they have any regulatory roles – especially wrt the duplicated HERV-K (C4) in the C4A and C4B? Are they structurally polymorphic in human populations? For an example of 4 dimorphic SVA in the MHC class I region see the paper by Kulski et al (Polymorphic SVA, Immunogenetics 2010, DOI 10.1007/s00251-010-0427-2). Do they generate functional haplotypes, i.e., does the C2 to TNXB region express or function differently between SVA negative and SVA positive haplotypes? Is this NSDK region more susceptible to recombinations or deletions because of the presence of two SVA and the adjoining HERV-K (C4)?

P13, 449-457. In section 5, the shark/zebrafish, frog and platypus studies all need to be referenced.

P13, 458. Prior to the conclusions, a section on how these 4 NSDK genes are regulated would be an useful addition, particularly if any of them are coordinated by the same transcription factors, or regulatory factors, cytokines, IFNs etc. Is there any evidence to indicate that the NSDK gene activities are coordinated as a single unit or function or act like an eukaryotic operon? Also, are there shared interactions of the NSDK genes with other genes within and outside the MHC genomic region?

P13, line 470. It should be pointed out in the conclusions that SNPs within the genes between C2 and SKI12W have been associated with age-related macular degeneration (AMD), although at least one or more SKI12W allelic variants seem to provide a protective effect against AMD.

P18, lines 479-480. It is unclear how the NSDK genes are involved in self-nonself recognition at the RNA levels. This needs some explanation; perhaps in its own section if there is sufficient evidence to support this hypothesis that was raised in the abstract (line 28) and the conclusion (line 479-480), but not discussed elsewhere.  

Author Response

Second reviewer:

Comments and Suggestions for Authors

The review entitled ‘An RNA Metabolism and Surveillance Quarter in the Major Histocompatibility Complex‘ updates some of the information about 4 interesting genes in the human MHC class III that were discovered in part by the co-author CY Yu and colleagues in the 1990s. This cluster of 4 adjoining genes NELFE, SKIV2L, DXO, STK19 is referred to as the NSDK multiple gene region. The review is interesting and important because it attempts to connect the regulation of self and nonself (viral infections) transcription and translation with some of the genes of the HLA class III region that are involved in the immunological RNA world. Immunologists when reviewing the overall structure, gene interactions and functions of the human MHC genomic region often neglect the details of this important gene region between C2 and C4.

In general, the review is clearly written and covers most publications and aspects of the NSDK gene region that have emerged during the last 25 years. However, there are a few points in the review that are unclear or have been intentionally or accidently omitted, but that could be reconsidered or added for a clearer and more comprehensive evaluation of the role and interactions of the NSDK genes.

It is not clear whether the NSDK gene region is an operon regulated by similar or different promoters and enhancers. Some conclusion or speculation could be made in regard to this possibility. Is the NSDK gene region a convenient designation simply based on their proximity or is it an example of an eukaryotic operon?

Response: There is no data suggesting the possibility of an “operon-like” mechanism for the expression of the NSDK genes. At the end of the manuscript, we did raise the question about concerted controls or responses of NSDK genes during growth and development (line 542).

What are the quotes in italics beneath the headings and where do they come from (Ref?)? Eg., p 2, lines 60-61; p 6, lines 217-219; p 9, lines 293 – 294; p 11. Lines 363;

Response: The quotes come from the authors who made attempts to introduce new and complex biologic concepts.

Figs 1 and 3 are too small; difficult to read the text within figs, eg for Fig 1 Alu, numbers, whatever is written in the helicase domain, etc , impossible to read without magnification. This problem applies to some of the other figures as well.

Response: We have increased the size of figures.

Table 1 needs references or a reference column.

Response: NCBI websites are added to the four genes described in Table 1.

P2, Fig 1A. if possible, please add and label the locations of the important sequences ZBTB12 and C2-AS1 within the C2 gene structure.

Response: The 5’ end of the C2 gene requires meticulous experimental characterization. In the legend of figure 1, we added a note for the location of ZBTB12 and C2-AS1.

P2, line 64. The CFB in the text is Bf in Fig 1. Consistency with terminology required.

Response: We have chosen to use Bf as the gene symbol for complement factor B throughout the manuscript.

P2, line 65. What transcription factors bind to the 5’ regulatory region of the RD gene?

Response: This requires in-depth investigations in future.

P3, line 92. What types of RNA are known to bind to the RRM (mention here)?

Response: It was reported that DSIF and NELF bindings require a nascent transcript longer than 18 nt to stably associate with the RNA Polymerase II elongation complex. (https://doi.org/10.1073/pnas.1000681107).

P3, line 98. What chromosome is DRB (DSIF) on?

Response: DSIF elongation factor have its subunits SUPT5H and SUPT4H1 located on chromosome 19 and chromosome 17 respectively.

P4, line 105. Is this NFLF-E or should it be NELF-E? What is the difference if they are different?

Response: The typographical mistake has been corrected, thanks.

P4, line 107. What NELF genes produce the A, B subunits and where are they relative to NELF-E gene? Please indicate where the NELF-C-D gene is located. Is their expression coordinated? What co-ordinates their expression?

Responses: NELFA gene is located on chromosome 4; NELFB gene is located chromosome 9; and NELFC/D gene is located chromosome 20. We do not known whether or how the expression of the four subunits of NELF are coordinated.

P4, line 111. Delete ‘the’

Response: Done as suggested, thanks.

P4, line 131. What is ARS2? This has suddenly appeared from nowhere with no prior explanation. Same with PHAX? Maybe a Table listing the interacting proteins, genes and gene locations would be an aid to a better understanding of the function(s) of NELF-E.

Responses: Thanks for the suggestion. We have gone over the manuscript thoroughly and defined all abbreviations. The NCBI websites for NELF-E and the other three genes have been added to Table 1. The NCBI site for NELF-E has a section describing specific proteins interacting with NELF-E.

P5, Fig. 2. Why is EC starred * (EC*) in the text of the figure?

Response: The EC* stands for “active elongation complex” in the cryo-EM structure.

P6, line 193-194. NELF suppression results in overexpression of HepC virus and this also may be related to hepatocellular carcinoma (Iida M, et al. Oncol Rep, 2012 Aug. PMID 22614758). Also, NELFE might regulate HIV transcription as described in the text and NELFE knockdown increases HIV transcription (Rao JN, et al. Biochem J, 2006 Dec 15. PMID 16898873).

Response: Thanks for the information. Reference cited.

P6, 198. miR-1236 is a negative regulator of VEGFR-3 signaling (Jones, Li et al 2012 ATVB, doi: 10.1161/ATVBAHA.111.243576). This is interesting in relation to the treatment of macular degeneration (MD) and cancers with antiVEGF therapy ( PMID 26857947; doi:10.1101/cshperspect.a006577)and raises a possible role of miR-1236 in the treatment or regulation of MD. miR-1236 can reduce hepB virus replication and protein production (Huang et al 2016, doi: 10.1038/srep34740)

Response: Thanks for the information.

P6, line 210. Grammar. ZEB1 is an … inducer (not inducers).

Response: Done as suggested. Thanks.

P8, Table 2. GWAS also have implicated C2/C2-AS1, C3, CFB (doi: 10.1073/pnas.0912702107) and VEGFA (Yu et al 2011, doi: 10.1093/hmg/ddr270) in age-related macular degeneration (AMD). Are the high risk SNP of these genes in LD? Are there haplotypes linking the AMD SNP of C2/C2-ASI, CFB, NELFE, SKIV2L in the class III region with VEGFA?

Response: We respectfully point out that AMD is not a focus of this manuscript.

P 11, 375. Delete an ‘is’ and express as ‘RP1 is expressed ubiquitously in ..’

Response: The sentence was revised. Thanks.

P12, 395 – 399. The two SVAs located in the C2 and the STK19 genes are only about 30 kb apart (Fig. 1). Do they have any regulatory roles – especially wrt the duplicated HERV-K (C4) in the C4A and C4B? Are they structurally polymorphic in human populations? For an example of 4 dimorphic SVA in the MHC class I region see the paper by Kulski et al (Polymorphic SVA, Immunogenetics 2010, DOI 10.1007/s00251-010-0427-2). Do they generate functional haplotypes, i.e., does the C2 to TNXB region express or function differently between SVA negative and SVA positive haplotypes? Is this NSDK region more susceptible to recombinations or deletions because of the presence of two SVA and the adjoining HERV-K (C4)?

Response: These are big questions that deserve future investigations…these will be topics for chapter 2!

P13, 449-457. In section 5, the shark/zebrafish, frog and platypus studies all need to be referenced.

Response: We have added URL sites. Thanks.

P13, 458. Prior to the conclusions, a section on how these 4 NSDK genes are regulated would be an useful addition, particularly if any of them are coordinated by the same transcription factors, or regulatory factors, cytokines, IFNs etc. Is there any evidence to indicate that the NSDK gene activities are coordinated as a single unit or function or act like an eukaryotic operon? Also, are there shared interactions of the NSDK genes with other genes within and outside the MHC genomic region?

Response: It’s one of our intents for this review to stimulate interests for future concerted investigations of the NSDK genes.

P13, line 470. It should be pointed out in the conclusions that SNPs within the genes between C2 and SKI12W have been associated with age-related macular degeneration (AMD), although at least one or more SKI12W allelic variants seem to provide a protective effect against AMD.

Responses: It is pretty well-established that variants of complement proteins in the alternative complement pathway are engaged in the pathogenesis of aged macular degeneration. The factor B gene is located in the HLA and its polymorphisms are in linkage disequilibrium with those of C2, C4A and C4B and HLA class I and class II genes. It would not be surprising to have polymorphic variants of SKIV2L (or SKI2W) forming extended haplotypes with HLA class I, II and III genes spanning thousands of kilobases. In this review we attempt to focus on the NSDK genes located between factor B and C4. It seems parallel efforts on the complement, and HLA class I and class II genetic variants would be desirable.  

P18, lines 479-480. It is unclear how the NSDK genes are involved in self-nonself recognition at the RNA levels. This needs some explanation; perhaps in its own section if there is sufficient evidence to support this hypothesis that was raised in the abstract (line 28) and the conclusion (line 479-480), but not discussed elsewhere.

Responses: On describing the functions of the four genes NSDK, we attempted to establish their roles on the detection and controls of intact and aberrant RNA molecules through their structures and integrity at the 5’ ends (or caps) and 3’ ends (or tails). Discrimination of self and non-self (host and virus) at the RNA level is an emerging and evolving concept.

Reviewer 3 Report

This review manuscript addresses four gene cluster residing in the middle of MHC class III region between complement C2/Bf and C4 genes. Authors discribe the genomic organization and function of each genes in great details with clinical implication. From evolutionary point, it is interesting how these genes linked together in the middle of the complement genes, but their involvement of RNA metabolism and surveillance is interesting. The review is written well in general except evolution section is problematic and needs to be carefully rewritten.

1)      Lines 444-457:

In general, this section lacks references. For example, authors state “The linkage of all four genes together as a group with MHC class II genes seems to begin in frogs”. (line 352-453) data source need to be sited here (i.e. Either reference Nature (2016) 538:336-343 or data source URL).

Lines 449-450: “The physical linkage of NELFE and SKIV2L, in one chromosome, and DXO and STK19 in another chromosome are observed in shark or zebrafish”. Without data sources, I cannot assess the facts. But looking at the NCBI databases, this statement is only partially supported.  In the NCBI databases, no-linkage status is true for zebrafish chromosomes (3 and 11), but linkage status in shark is not known. Because they are found in different scaffolds, that does not mean that they are in different chromosomes. Considering shark MHC has not been assembled so far, all four genes may likely to be on the same chromosome. Bony fish genome has been known for the disrupted genome; MHC class I, II, III genes are found in separate chromosomes. Also note that DXO and STK19 genes in zebrafish are 4Mbp apart, not next to each other like other species including sharks, suggesting genome modification in zebrafish (likely all bony fish). Authors may need to look at gar (basal bony fish) and Coelacanth (close to tetrapod) for better understanding linkage status.   

Line 444-448: Provide a figure explaining hypothetical scenario of evolutionary history of NELFE/RD genes. It is difficult to grasp from a few species mentioned, but a figure will help readers clarify authors points.

2)      Figure 1A: color is not showing well. 

Figure 1B: B is lost at the bottom of 1A

3)      Figure 3 needs magnifications.

Author Response

The third reviewer:

Comments and Suggestions for Authors

This review manuscript addresses four gene cluster residing in the middle of MHC class III region between complement C2/Bf and C4 genes. Authors describe the genomic organization and function of each genes in great details with clinical implication. From evolutionary point, it is interesting how these genes linked together in the middle of the complement genes, but their involvement of RNA metabolism and surveillance is interesting. The review is written well in general except evolution section is problematic and needs to be carefully rewritten.

1)     Lines 444-457:

In general, this section lacks references. For example, authors state “The linkage of all four genes together as a group with MHC class II genes seems to begin in frogs”. (line 352-453) data source need to be sited here (i.e. Either reference Nature (2016) 538:336-343 or data source URL).

Lines 449-450: “The physical linkage of NELFE and SKIV2L, in one chromosome, and DXO and STK19 in another chromosome are observed in shark or zebrafish”. Without data sources, I cannot assess the facts. But looking at the NCBI databases, this statement is only partially supported. In the NCBI databases, no-linkage status is true for zebrafish chromosomes (3 and 11), but linkage status in shark is not known. Because they are found in different scaffolds, that does not mean that they are in different chromosomes. Considering shark MHC has not been assembled so far, all four genes may likely to be on the same chromosome. Bony fish genome has been known for the disrupted genome; MHC class I, II, III genes are found in separate chromosomes. Also note that DXO and STK19 genes in zebrafish are 4Mbp apart, not next to each other like other species including sharks, suggesting genome modification in zebrafish (likely all bony fish). Authors may need to look at gar (basal bony fish) and Coelacanth (close to tetrapod) for better understanding linkage status.  

Response: Information about the primary structure of NELF-E with and without repetitive Arg-Asp (RD) sequences, the presence of gene pairs for NELF-E and SKIV2L, and DOX and STK19, and the emergence of the quartet in linkage were obtained by thorough analyses of gene maps from the NCBI websites for various species. The URL for the two gene pairs in representative species in vertebrates and mammals are included in the text. The Nature reference for frog genome has been cited.

Line 444-448: Provide a figure explaining hypothetical scenario of evolutionary history of NELFE/RD genes. It is difficult to grasp from a few species mentioned, but a figure will help readers clarify authors points.

Response: This is an excellent suggestion. We will make good efforts to create such a figure in a future publication.

2)     Figure 1A: color is not showing well. Figure 1B: B is lost at the bottom of 1A

Response: These will be corrected at the proof stage.

3)     Figure 3 needs magnifications.

Response: Photographs of HeLa cells were taken under ultraviolet microscope with 10x40 magnifications.

Round 2

Reviewer 2 Report

Query: P18, lines 479-480. It is unclear how the NSDK genes are involved in self-nonself recognition at the RNA levels. This needs some explanation; perhaps in its own section if there is sufficient evidence to support this hypothesis that was raised in the abstract (line 28) and the conclusion (line 479-480), but not discussed elsewhere.

Reply: On describing the functions of the four genes NSDK, we attempted to establish their roles on the detection and controls of intact and aberrant RNA molecules through their structures and integrity at the 5’ ends (or caps) and 3’ ends (or tails). Discrimination of self and non-self (host and virus) at the RNA level is an emerging and evolving concept.

The concept of self-nonself recognition and surveillance through the HLA molecules and the T/NK-cell system is well defined and accepted in immunology and the MHC system. On the other hand, the provocative new concept on how the NSDK plays a different level of self-nonself recognition and surveillance through RNA is not explained by the authors in this paper and therefore remains misleading as it stands and should be deleted from the manuscript. This unexplained concept of ‘Discrimination of self and non-self (host and virus) at the RNA level‘ could be developed and explained separately in a different paper or chapter.

Change ‘on the recognition of self and non-self’ on line 28 to ‘roles in health and disease at the RNA levels,’

Delete ‘NSDK seems to play a different level of self-nonself recognition and surveillance through RNA’ from line 810 and 811.

Please use ‘spell check’ to find and correct spelling errors, eg minmum (line 381), causesthe (line 480), noticeds (line 490), ubiguitously (line 758).

Author Response

The concept of self-nonself recognition and surveillance through the HLA molecules and the T/NK-cell system is well defined and accepted in immunology and the MHC system. On the other hand, the provocative new concept on how the NSDK plays a different level of self-nonself recognition and surveillance through RNA is not explained by the authors in this paper and therefore remains misleading as it stands and should be deleted from the manuscript. This unexplained concept of ‘Discrimination of self and non-self (host and virus) at the RNA level‘ could be developed and explained separately in a different paper or chapter.

Change ‘on the recognition of self and non-self’ on line 28 to ‘roles in health and disease at the RNA levels,’

Delete ‘NSDK seems to play a different level of self-nonself recognition and surveillance through RNA’ from line 810 and 811.

Please use ‘spell check’ to find and correct spelling errors, eg minmum (line 381), causesthe (line 480), noticeds (line 490), ubiguitously (line 758).

Responses:

We are very appreciative of the critiques. We have revised the manuscript extensively and removed the claim on "self-nonself recognition at the RNA level". 

The original idea was originated on the pattern recognition of viral or foreign RNA molecules without a mature m7G-cap or a poly-A tail, which are recognized by receptors for RIG1 and MDA5 (RLRs).  Skiv2l modulates the type 1 interferon response through RLRs during a viral infection.  Throughout manuscript including the abstract, the section on Skiv2l, and the conclusion we made efforts to clarify this point. The emphasis of the manuscript is focused on the roles of NSDK proteins on the metabolism and surveillance of RNA integrity and their disease associations.  

We have inactivated the "track changes" program, which is distractive, and corrected typographical errors in the manuscript.  Relevant changes made in this version are highlighted in yellow.

Thanks again for the constructive suggestions.

Reviewer 3 Report

I acknowledge authors efforts to improve manuscript, but Section 5 “evolution” (lines 739-759) is still poorly written and not logical in some parts.

Line 739-747: Some NSDK quartet genes are present in invertebrates (i.e. drosophila, C. elegance, yeast). It would be nice to examine what genes are in the vicinity of NSDK genes in these species.

Line 748-750: Authors changed from “different chromosome” to “different chromosomal fragment”. Statement is correct. However, the sentence ends abruptly and gives readers an impression that these chromosomal fragments are in different chromosomes. Please add a sentence something like ”these fragments are short and linkage status is not yet known”.

Line 752-754: Authors state frog NSDK quartet are linked to “class II” in frog genome. Since authors emphasis has been mapping NSDK quartet between complements C2/Bf and C4A/B, it is better to stick with the theme. My suggestion is to use “class III” or “complements” instead of “class II”.

Line 754: “could have emerged earlier. In egg-laying mammal platypus…” This part reads egg-laying platypus is evolutionary older than amphibian frogs. Amphibians diverged ~350million years ago, while monotreme diverged ~220 million year ago. Rewrite this section (example: removing “but such physical linkage could have emerged earlier”).

Author Response

We are grateful to the reviewer for the useful comments on the evolution of the NSDK genes. Specific changes in response to the critiques are listed as follows.

We have gone over the locations of the NELF-E, SKIV2L, DOX and STK19 genes in genomes of invertebrates. There are no evidence for any physical linkage among these four genes.  Therefore, we add the following sentence to state this phenomenon - lines 548-549:

"There is no evidence for a physical linkage for any two genes of the NSDK quartet prior to vertebrates."

About the physical linkage of NELF-E, SKIV2L, DOX AND STK19, we followed the suggestion of the reviewer to add the following sentence at lines 550-551:

"For the sequence data deposited in the NCBI, most chromosomal segments are relatively short and their linkage status is not yet known." 

We have adopted the suggestion by the Reviewer to remove the phase "but such physical linkage could have emerged earlier." - lines 556-557.

Thanks again for the suggestion.

We have also made corrections on typographical errors. Relevant changes are highlighted in yellow.